# Knowledge graph empowerment from knowledge learning to graduation requirements achievement

Yangrui Yang[1], Sisi Chen [1]*, Yaping Zhu[1], Hao Zhu[2], Zhigang Chen[2]

**1** Dept. of Information Engineering, North China University of Water Resources and Electric Power, Zhengzhou, Henan, China, **2** Dept. of Informatization Office, North China University of Water Resources and Electric Power, Zhengzhou, Henan, China

* chensisi@stu.ncwu.edu.cn

## Abstract

A deep understanding of the relationship between the knowledge acquired and the graduation requirements is essential for students to precisely meet the graduation requirements and to become human resources with specific knowledge, skills and professionalism. In this paper, we define the ontology layer of the knowledge graph by deeply analyzing the relationship between graduation requirement, course and knowledge. Based on the implementation of the concept of Outcome Based Education, we use Knowledge extraction, fusion, reasoning techniques to construct a hierarchical knowledge graph with the main line of "knowledge-course-graduation requirements. In the process of knowledge extraction, in order to alleviate the huge labor overhead brought by traditional extraction methods, this paper adopts a transfer learning method to extract triadic knowledge using the multi-task framework EERJE, Finally, knowledge reasoning was also performed with the help of LLM to further expand the knowledge scope. The comprehensiveness, correctness and relatedness of the data were evaluated through the experiment, and the F1 value of the ternary group extraction was 87.76%, the accuracy rate of entity classification was 85.42%, the data coverage was more comprehensive, and the results showed that the data quality was better, and the knowledge graph constructed in this way can fully optimize the organization and management of teaching resources, help students intuitively and comprehensively grasp the correlation and difference between graduation requirements and various knowledge points, and let the Students can carry out personalized independent learning through the navigation mode of knowledge graph, strengthen their weak links, and complete the relevant graduation requirements, which effectively improves the degree of students' graduation requirements achievement. This new paradigm of knowledge graph enabled teaching is of reference significance for engineering education majors to improve the degree of graduation requirements achievement.

**Data Availability Statement:** Data is available at https://github.com/aichiroudemao/database.

**Funding:** This research was funded by the Henan Provincial Higher Education Teaching Reform Research and Practice Project (2021SJGLX017),

and the North China University of Water Resources and Electric Power Education Reform Project. They played a role in the decision to publish this step.

**Competing interests:** The authors declare no competing interests.

## Introduction

Knowledge are an essential foundational component in the educational and teaching process, and the learning of knowledge is usually accomplished through specific learning activities. At the same time, the design of learning activities is intended to achieve graduation requirement indicators. The graduation requirement attainment system in colleges and universities requires that each course and each knowledge unit closely supports graduation requirements and comprehensively covers graduation requirements [1], and all specific knowledge points together guide students from knowledge learning to graduation requirement attainment. knowledge of existing syllabi, instructional design, and web pages about training programs are usually presented independently in the form of unstructured texts. Redundant descriptions and information overload make it easy for students to miss key effective information and lead to unclear perception of the connection between knowledge and graduation requirements, which may prevent students from understanding the practical use and value of knowledge in the learning process and make it difficult to clarify the specific steps and ways to achieve graduation requirements. For example, when engineering management students learn the knowledge point of "original voucher" in the course of "Accounting", if they do not know the graduation requirement of "engineering knowledge", they may only superficially understand that "original voucher" is the initial source of recording financial transactions, but they cannot relate it to the cost and quality of engineering, so they cannot precisely and deeply reach the graduation requirement of "engineering knowledge". For example, when engineering management students learn the knowledge point of "original voucher" in the course of "Accounting", if they do not know the graduation requirement of "engineering knowledge", they may only superficially understand that "original voucher" is the initial source of recording financial transactions, but they cannot relate it to the cost and quality of engineering, so they cannot precisely and deeply reach the graduation requirement of "engineering knowledge". Therefore, it is particularly urgent and necessary to comprehensively construct the connection between knowledge units and graduation requirements.

The UN Summit on Transforming Education recently released the Initiative for Action to Ensure and Improve the Quality Public Digital Learning for All [2], which states that countries can take full advantage of digital technologies to empower teaching and learning. As a new generation of information technology, knowledge graph is a powerful tool for computer-supported knowledge organization and visual presentation, Existing generative AI has randomness in answering questions due to the massive breadth of the training data, which makes it very easy to generate answers that are not related to the current question and need to be restricted. In contrast, knowledge graphs can extract structured information from the target document, which in turn provides structured knowledge support for applications such as retrieval and Q&A, and knowledge graphs contain clear inter-entity relationships, which help machines better understanding of semantics, thus improving the precision and accuracy of information processing. In addition, for answering questions in vertical domains, generative AI may introduce other domain noises, and knowledge graph can provide regular and accurate professional information for downstream applications, which is easier to meet the specialized needs of vertical domains. Therefore, in order to help students understand the connection between various types of knowledge and graduation requirements more intuitively and comprehensively, This paper constructs a hierarchical knowledge graph with the main line of "knowledge-course-graduation requirements" to support students in transitioning from knowledge learning to the attainment of graduation goals. Specifically, this paper analyzes various teaching resources such as training programs and syllabi to clarify the specific ability requirements and knowledge requirements covered in graduation requirements. It breaks

down graduation requirement indicators, summarizes the relationship between graduation requirement, course, and knowledge, and defines the ontology layer of the knowledge graph. In the process of constructing the graph, entity relationship extraction is one of the key technologies. Entity relationship extraction is an information extraction method based on natural language processing and machine learning techniques, which can identify and extract entities and relationships from unstructured or semi-structured texts and represent them as structured data in the form of triplets and knowledge graphs. Existing extraction methods based on summary sentence types [3–5] or sequence annotations [6–8] are often accompanied by great labor overhead, and to alleviate this labor overhead, This article describes the use of the APICKnow [9] to define an Education Entity Relationship Joint Extractor(EERJE) for extracting knowledge triplets. Using the graph database Neo4j as the knowledge storage carrier, the triplet knowledge is presented in the form of a knowledge graph. The graph, by connecting various related entities, constructs a complete knowledge structure. This comprehensive structure can help students more intuitively understand the various abilities and knowledge required for graduation requirements, and can help them clarify their current learning progress and future learning goals, thereby enhancing their intrinsic motivation for learning. The knowledge graph can guide students to gradually master each knowledge unit to precisely achieve the graduation requirements, thus comprehensively improving students' degree of graduation goal attainment.

The main work of this paper includes:

- We analyze the content characteristics of heterogeneous data sources and define clear entity types and inter-entity relationship types based on the main line of "knowledge-course-graduation requirements" to build a knowledge graph ontology layer.

- We use EERJE, a joint entity-relationship extraction framework, to perform knowledge triad extraction and build a knowledge graph that supports learning from knowledge toward graduation requirements achievement.

- Utilizing LLM for entity relationship reasoning to further expand the scope of knowledge.

- Evaluate the quality and use effect of the graph through experiments, and the results show that the graph allows students to clearly understand the links and differences between the various knowledge points that support graduation requirements, facilitating knowledge sorting and learning, while students can use the content of the knowledge graph according to their own learning mastery, jumping to the mastery of the knowledge of the weak knowledge points to learn, so that they can be personalized and independent learning, so that students can efficiently and accurately complete the relevant graduation requirements, and thus improve the degree of achievement of graduation requirements. This enables students to complete the graduation requirements efficiently and accurately, thus improving the degree of fulfillment of graduation requirements.

## Related work

The development of the knowledge graph concept has gone through the semantic web [10]-ontology-world wide web [11]-semantic web [12]-linked data [13]. It wasn't until May 2012 that Google officially introduced and published the concept of the Knowledge Graph [14], which is a general semantic knowledge formalization framework. It primarily uses a graph-based data model to capture knowledge in application scenarios involving integration [15], and employs visualization techniques to describe knowledge resources and their carriers,

mine, analyze, construct, draw, and display knowledge and their interconnections, providing an efficient way to organize, manage, and analyze massive amounts of data [16], and Knowledge graph has become an essential tool for semantic analysis with the development of natural language processing and deep learning. A high-quality knowledge graph is handy for building a high-performance knowledge-driven application [17], so knowledge graphs are being applied to various fields of life as a highly regarded hot technology, and have shown its great application prospects in all walks of life.

Since knowledge graphs as large-scale knowledge engineering are crucial for the organization and integration of knowledge [18], entity-relationship extraction techniques can be used to integrate the data in the form of entities, inter-entity relationships, and attributes of entities extracted from unstructured or semi-structured text data through triads, such as <head entity, relationship, tail entity>, <entity, attribute, attribute value>, so that fragmented and abstract knowledge can be displayed clearly and visually. In the field of education alone, several scholars have conducted corresponding researches using knowledge graphs. For example, Chen et al. [19] extract concepts of subjects or courses, and then identify educational relationships among the concepts. Ding et al. [20] analyzed the research hotspots and evolution trends in the field of intelligent education research. Chen et al. [21] conducted an in-depth analysis of the mathematical discipline.

Although the aforementioned studies targeting the field of education have been widely applied in terms of subject concepts, teaching research hotspots, and the discipline of mathematics, and have utilized related knowledge to construct graphs, no one has yet effectively organized the related contents corresponding to graduation requirements. In order to help teachers and students better understand the content of their graduation requirements, we will make a specific study for this. In this paper, we will deeply analyze the relevant text data, realize the association between different entities through the fusion of multi-source heterogeneous data, build a knowledge graph with graduation requirements as the core, It makes the correlation and difference between each knowledge point and the graduation requirements more intuitively displayed, helps students to better understand the specific knowledge points corresponding to graduation requirements, and to reach the graduation requirements step by step according to the content contained in the knowledge graph, in order to improve the degree of graduation requirement achievement.

## Knowledge system description

Based on the OBE concept, this paper analyzes the relevant text data in depth and summarizes the main structure of "Knowledge-Course-Graduation Requirements" to help students learn the knowledge points, complete the corresponding course objectives, and acquire knowledge, skills and professionalism to meet the graduation requirements. The types and definitions of each level are shown in Table 1.

## Knowledge graph structure

The data sources of the knowledge graph constructed in this paper are syllabus, teaching design, web knowledge about cultivation program, etc., mainly unstructured and semi-structured text corpus, and the graph constructed in this paper belongs to vertical domain knowledge graph according to the division of knowledge coverage and knowledge mining depth. Vertical domain knowledge graphs are oriented to fixed domains, and the degree of mining knowledge in fixed domains is deeper and requires higher quality data. From the above two aspects, this paper will adopt the top-down approach to construct the knowledge graph, and the overall construction process is shown in Fig 1. First, the web knowledge data of syllabus,

**Table 1. Types included in each main line structure and related descriptions.**

| Structure Level | Type included | Related Description |
|---|---|---|
| Knowledge | Knowledge Units | A collection of knowledge points |
| | Specific knowledge points | Specific knowledge points around the course |
| Course | Course Title | Course Title |
| | Course Objectives | Describe the objectives to be achieved upon completion of the course |
| | Author | Authors of selected teaching materials |
| | Prerequisite Courses | Subjects that should be taken prior to studying the current course |
| | Selected teaching materials | List of books used in the course lectures |
| | Publisher | Publisher of selected materials |
| | Labs | Related experiments around the course |
| Graduation Requirements | Engineering Knowledge | Ability to use fundamental and specialized knowledge to solve complex engineering problems |
| | Problem Analysis | Be able to apply knowledge to analyze complex engineering problems in order to obtain valid theories. |
| | Design/Develop solutions | Ability to design/develop solutions to complex engineering problems |
| | Research | Ability to study complex engineering problems based on scientific principles and methods |
| | Use of modern tools | Ability to use modern engineering tools and information technology tools to solve complex engineering problems |
| | Engineering and Society | Ability to perform sound analysis and understand responsibilities based on engineering related background knowledge |
| | Environment and Sustainable Development | Be able to understand and evaluate the impact of engineering practice on environmental and social sustainability. |
| | Professional Standards | Ability to understand and adhere to engineering ethics and codes of practice and fulfill responsibilities in engineering practice |
| | Individual and Team | Ability to assume the role of individual, team member and leader of a team in a multidisciplinary context |
| | Communication | Ability to communicate effectively with industry peers and the public on complex engineering issues |
| | Project Management | Understand and master engineering management principles and economic decision-making methods and apply them |
| | Lifelong Learning | Ability to continuously learn and adapt to developments |

teachers' lesson plans, and relevant training programs should be obtained, and the obtained data should be redundantly filtered to remove paragraphs that are irrelevant to graduation requirements, course objectives, and knowledge units, and the filtered paragraphs should be divided into sentences using PyLTP [22] with a period as a separator, and the ontology structure of the knowledge graph should be constructed on this basis, which is used for the graduation requirement-related content modeling and specification; second, the manual annotated text data is used to train the federated extraction framework EERJE in order to extract entities and inter-entity relationships from other text data; then, the unstructured and semi-structured text data are fused for multi-word one-sense problems using techniques such as entity linking, and the structured text data are merged with relational databases; Then use LLM for knowledge reasoning to further expand the data range; finally, the graph database is used to The more popular Neo4j database [23] is used to store the data layer of the knowledge graph.

## Ontology frame design

The most important aspect of the construction of the knowledge graph is the design of the ontology, which can formally represent the hierarchical relationships between entities [24].

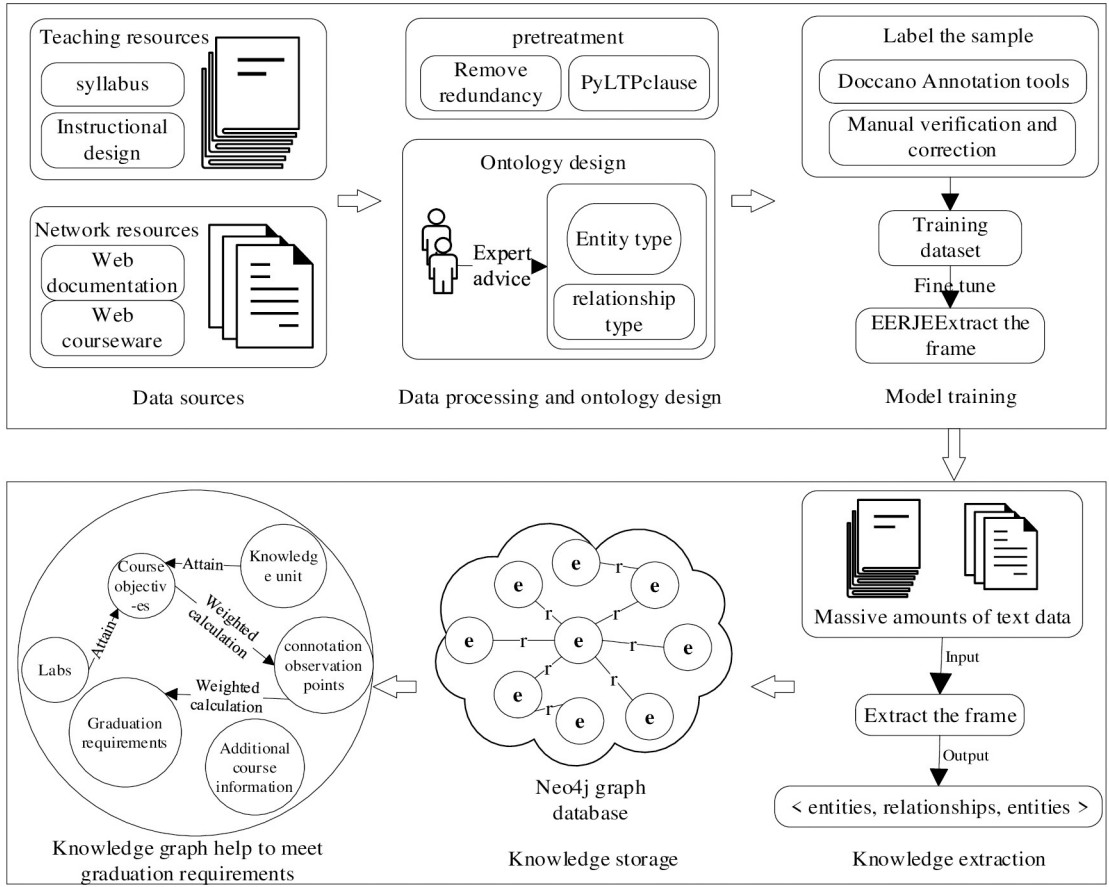

**Fig 1. Overall build process.**

The design of the ontology structure determines whether the graph can facilitate the application in this domain. Therefore, this paper defines a total of 10 entity types and 11 relationship types based on the description of the knowledge system and under the guidance of professional teachers, The ontology frame design is shown in Fig 2.

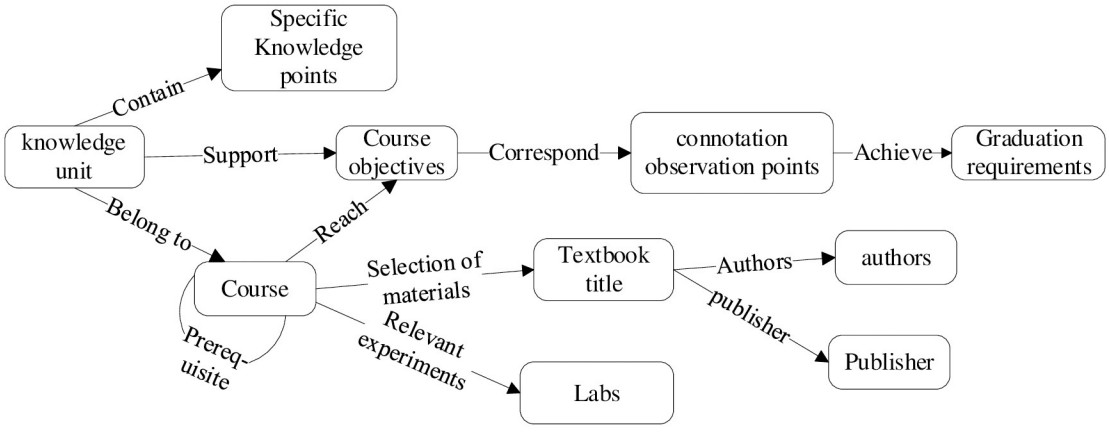

**Fig 2. Ontology frame design.**

## Knowledge extraction

**Data processing.** In this paper, we mainly use publicly available syllabi, teachers' lesson plans, and the content of web pages about training programs as data sources. After obtaining the relevant data, the first step is to filter out the redundant descriptions that are not related to graduation objectives, course system and knowledge units by paragraph screening, and then use PyLTP to process the filtered paragraphs with periods as separators in separate sentences. After manual annotation by hand, the processed text data are input to the joint extraction framework EERJE for training in order to extract entities and inter-entity relationships.

**Entity relationship extraction.** The overall processing flow of EERJE, the multitasking framework used in this paper, is a dynamic hint generator combined with a predefined schema to generate dynamic hint sequences for the input text, which in turn leads to a joint entity-relationship extractor to extract both entities and inter-entity relationships from the current input text. The overall processing flow is shown in Fig 3.

Multitasking framework EERJE can be divided into three parts related to the introduction: Dynamic prompt generator; Dynamic prompt sequences; Entity-Relationship Joint Extractor.

*I Dynamic prompt generator.* The kernel of the dynamic prompt generator is a Bert-based text classifier [25], which generates a list of candidate relations by inputting text to predict the types of relations that may exist in an utterance. However, it is worth noting that the number of this list of candidate relations should not exceed three, because the entity-relationship joint extractor extracts multiple entity-relationship triples present in a sentence based on the dynamic prompt sequence generated by the Dynamic prompt generator. In general, although a statement does not contain all relationship types at the same time, the entity-relationship joint extractor receives more noise as the number of defined relationship types increases. To avoid interference, let the dynamic prompt generator go ahead and narrow down the hints by outputting only the top three most likely relationship types.

*II Dynamic prompt sequence.* The dynamic prompt sequence is composed of entity types and relationship types combined with the current input text data in the form of [spot], [asso], [text], where [spot] refers to the entity type, [asso] refers to the relationship between entities, and [text] refers to the input text.

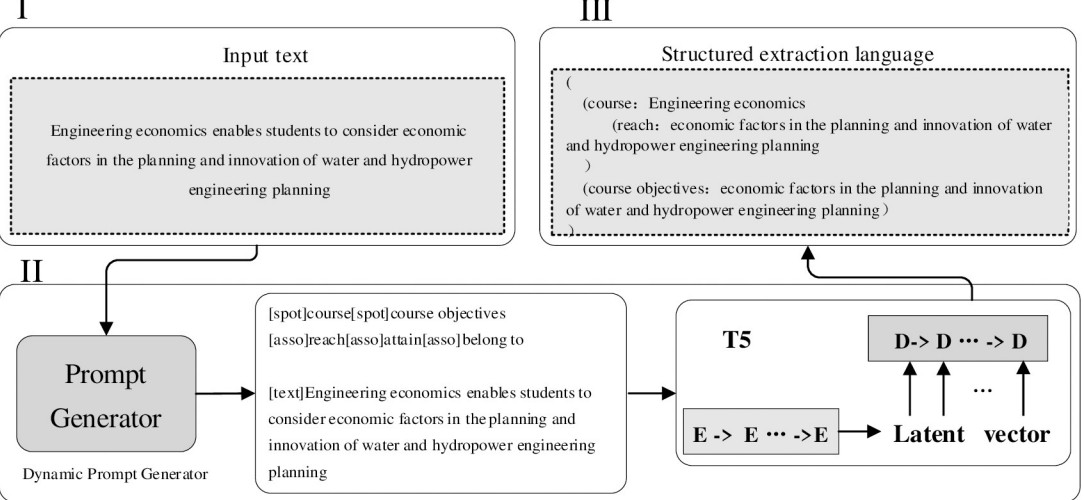

**Fig 3. EERJE overall processing flow.**

**Table 2. Entity relationship extraction results.**

| Entity Type 1 | Relationship Type | Entity Type 2 | Number of triplet strips |
|---|---|---|---|
| graduation requirements indicator points | achieve | graduation requirements | 96条 |
| course objectives | correspond | graduation requirements indicator points | 167条 |
| knowledge units | support | course objectives | 693条 |
| course title | reach | course objectives | 245条 |
| knowledge units | belong to | course title | 752条 |
| textbook title | publisher | publisher | 125条 |
| textbook title | author | author | 131条 |
| course title | relevant experiments | labs | 145条 |
| course title | selection of materials | textbook title | 263条 |
| knowledge units | contain | specific knowledge points | 843条 |
| course title | Prerequisite courses | course title | 426条 |

*III Entity-Relationship Joint Extractor*. The Entity-Relationship Joint Extractor is a sequence-to-sequence framework and the overall generation process can be expressed by Eq (1):

$$ERJE\left(\left[I_1, I_2, \ldots\ldots, I_{|I|}\right]\right) = \left[y_1, y_2, \ldots\ldots, y_{|y|}\right] \quad (1)$$

Among them, ERJE refers to entity relationship joint extractor, which is implemented based on T5 [26] of Transformer [27], $[I_1, I_2, \ldots\ldots, I_{|I|}]$ denotes dynamic prompt sequence, and $[I_1, I_2, \ldots\ldots, I_{|I|}]$ denotes generated sequence, i.e., the dynamic prompt sequence $I$ generated by dynamic prompt generator is input to T5, and the sequence $Y$ is generated by T5 to obtain the triadic knowledge of entities as well as inter-entity relationships.

*Knowledge extraction results*. In this paper, we take texts such as syllabi, instructional designs, and webpage knowledge data about training programs as the extraction corpus, and use EERJE to extract entity-relationship triples from this corpus. The cumulative number of entity relationship triples extracted is shown in Table 2.

## Knowledge fusion

In order to improve the quality of knowledge base construction [28], knowledge fusion of extracted entities is required, mainly to solve two problems: I. Unifying the representation of the same entity type in heterogeneous texts from multiple sources: Due to the interference of objective factors when different data sources describe the same entity, an entity may have multiple representations, which is the phenomenon of multiple words with one meaning. It is necessary to perform coreference resolution to standardize the entity representation. For example, during the extraction of input text data, the entity "Engineering Management major" is extracted from the teaching outline, while the entity "EngMgmt" is extracted from the teacher's lesson plan. These two entities represent different expressions for the "Engineering Management major" and need to be merged through a fusion operation. To address this phenomenon, machine learning algorithms can be employed to merge entities, build models using training data, and determine whether two entities can be fused based on the similarity of their attributes. Compared to traditional rule-based or similarity-based methods, using machine learning algorithms for entity merging does not require manual setting of rules or weights. II. Resolving entity ambiguity issues. In data sources, an entity's representation may have two different meanings, which is the phenomenon of one word with multiple meanings, necessitating

entity disambiguation. For example, "Introduction to Networks" can refer to both " Computer Networks " and " Social Networks ". These two interpretations of "Introduction to Networks" represent different subject areas—one in computer science and the other in social sciences. To address this one word with multiple meanings phenomenon, we can first provide a list of target entities and set rules to filter out unlikely target entities. This process helps identify candidate entities. Then, an entity linking method is employed to determine the true target entity that a mention refers to within the candidate set [29].

## Knowledge reasoning

In order to further expand the scope of the data, this paper uses the knowledge graph constructed with graduation requirements as the neural knowledge base, and lets a large-scale language model (e.g., GPT3.5) learn the knowledge contained in the knowledge graph and reason out more entity relationships related to the hierarchy of "Knowledge-Course-Graduation Requirements" through three modules. The three modules are: Entity Extractor, Entity Knowledge Parser and Entity Relationship Decision.

## Entity extractor module

The Entity extractor module consists of entity extraction unit and entity pairing unit. That is, given a text, first with the help of the LLM extraction of the 10 types of entities defined in this paper, and then two and two pairs to form (entity_1, entity_2) form, and finally output entity pairs. The extraction case and output are shown in Fig 4.

**Entity extraction unit**

Entities related to graduation requirements, graduation requirements indicator points, course objectives, course title, textbook title, author, publisher, labs, knowledge units, and specific knowledge points were extracted from the text.

**Input：**
*Engineering Economics* focuses on indicators for evaluating economic effects, financial analysis and investment estimation of engineering projects.
**Output：**
Engineering Economics;
evaluating economic effects;
financial analysis and investment estimation of engineering projects

**Entity pairing unit**

Entities are matched in pairs

**Input：**
Engineering Economics;
evaluating economic effects;
financial analysis and investment estimation of engineering projects
**Output：**
(Engineering Economics, evaluating economic effects)
(Engineering Economics, financial analysis and investment estimation of engineering projects)
(evaluating economic effects, financial analysis and investment estimation of engineering projects)

**Fig 4. Entity extractor module.**

## Entity knowledge parser module

The Entity knowledge parser module consists of an Entity Knowledge Mining unit and an Entity Knowledge Paring unit. That is, given an entity pair, LLM mines the knowledge related to each entity in the entity pair, and then combines the two pieces of knowledge into a complete and integrated set of paragraphs. Considering that the module needs to reason about 11 types of relationships, a separate knowledge parser is designed for each relationship type to accomplish the related knowledge mining work through a parallel approach. In addition, in order to better mine the knowledge related to the relationship types, take the relationship type "belongs to" as an example, the prompts are also designed as "what knowledge units does the entity contain" and "in which courses does the entity appear? ". The other relationship types are the same as above, replaced with the relevant prompts. Example input and output as shown in Fig 5.

## Entity relationship decision module

The Entity relationship decision module consists of three relationship decision units and a result combination unit. That is, given entity pairs and related knowledge, the three decision units independently evaluate inter-entity relationships from different perspectives and styles (direct questioning, judging statements to be true or false, and formulating options), and then the result combination unit aggregates the outputs of the three decision units to finalize the inter-entity relationships through majority voting. In order to improve the accuracy, separate relationship decision makers are still designed for the 11 relationship types for parallel operation. In addition, due to the polysemous nature of these 11 relationship types, a single word may cause ambiguity and lead to invalid responses generated by LLM. For example, if the relationship type "belongs to" is associated with different knowledge in different domains, LLM may generate an ambiguous and invalid answer such as "they have a certain relationship with each other" instead of making a "yes" or "no" answer. " or "No" instead of making a direct

**Fig 5. Entity knowledge parser module.**

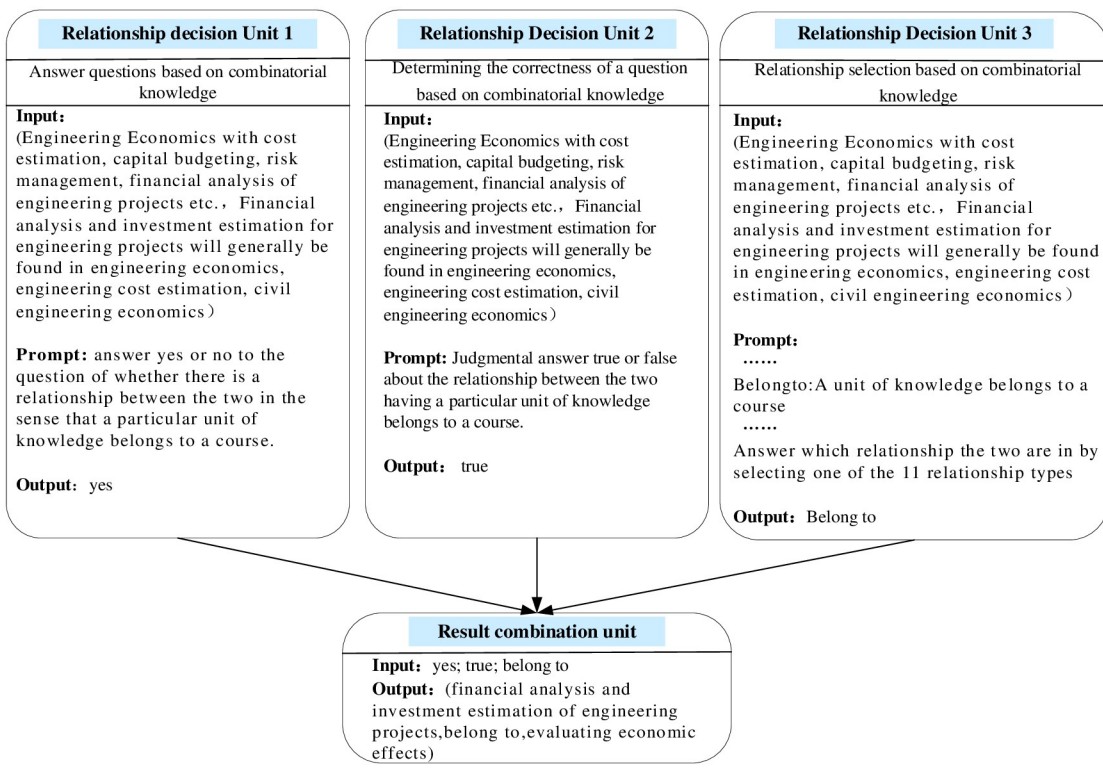

**Fig 6. Entity relationship decision module.**

judgment. For this reason, this paper replaces each relationship with a description of that relationship in each unit prompt, e.g., "belongs to" can be described as "a knowledge unit belongs to a course". Other relationship types as above need to be replaced with specific explanations. The specific output is shown in Fig 6.

## Graph database storage

Neo4j uses a graphical data model, has good scalability, can handle complex relational data and massive data, and its use of nodes and edges to represent the data, this storage structure and the real world of relational description is very close to people in the use of Neo4j query related information will be quite convenient. In addition, Neo4j supports complex relational queries, full-text indexing and advanced search, making it easier to query and analyze the knowledge graph. Considering that the later expansion and application of the knowledge graph require complex semantic search and also storage of large amount of data, etc., this paper will use Neo4j for data layer storage.

The entity relationship triad extracted by the federated extraction framework EERJE is imported into Neo4j. As an example, many entities and their relationships, such as graduation requirements, graduation requirement index points, course names, prerequisite courses, course objectives, textbook names, experiment publishers, etc. related to engineering management majors are shown in Fig 7 to demonstrate the specific storage form. The major consists of the following courses: Engineering Economics, Engineering Cost, Introduction to Engineering Graphics B, and Engineering Project Management. Among them, Engineering Project Management is a prerequisite course for both Engineering Economics and Engineering Cost.

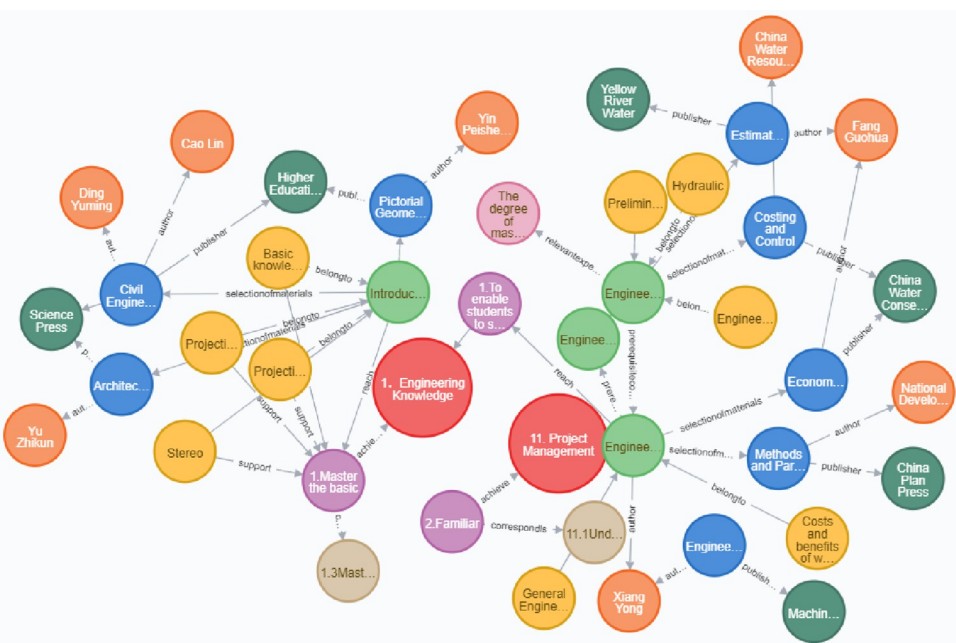

**Fig 7. Engineering management professional entity relationship storage form.**

Similar relationships exist among other courses. By establishing such prerequisite relationships, connections between courses are established [30]. In addition, graduation requirements are linked to graduation requirement index points, graduation requirement index points are linked to course names, course names are linked to textbook names, experiments, knowledge units, etc. By connecting entities to relationships and relationships to entities, the relevant content of the entire program is related, forming a complete structural system from specific knowledge points to graduation requirements.

## Knowledge graph evaluation and analysis

In order to construct a high standard knowledge graph, it is necessary to assess its data quality and usage effect; for the assessment of quality, Chen et al. [31] had developed a practical framework for the quality assessment of knowledge graphs, which summarizes the detailed basic quality requirements for each of the five common categories of knowledge graph-based applications, and each category of applications has different evaluation indexes. As for the quality assessment of vertical domain knowledge graphs, Nguyen et al. [32] proposed a detailed assessment of data quality, mainly assessing its correctness, relatedness, and comprehensiveness; in order to validate the quality of the knowledge graph constructed in this paper, it will be evaluated with the help of the three dimensions of correctness, relatedness, and comprehensiveness developed by Nguyen et al. For the assessment of the effectiveness of its use, this paper will use a questionnaire to allow students to evaluate themselves in order to determine whether the knowledge graph allows students to personalize their learning in order to improve the degree of graduation requirement attainment.

### Assessment of data quality

The assessment of data quality is mainly done in terms of correctness, relatedness, and comprehensiveness, where correctness means that the extracted entities/triples are accurate,

relatedness means that the extracted entities should be related to the defined types, and comprehensiveness means the information covered by the constructed graph.

For the assessment of correctness, this paper takes the manually labeled initial data and divides the initial text into training set, validation set and test set according to the ratio of 8:1:1 in accordance with the commonly used methods of corpus segmentation, and conducts labeling test for all kinds of relation triples in each utterance, and adopts the values of P, R, and F1 as the indicators of the model extraction performance assessment, and the formula of each indicator is as follows:

$$P = \frac{TP}{TP + FP} \times 100\% \tag{2}$$

$$R = \frac{TP}{TP + FN} \times 100\% \tag{3}$$

$$F1 = \frac{2}{TP + FN} \times 100\% \tag{4}$$

In the above formula, TP denotes the positive sample predicted to be a positive case, FP denotes the negative sample predicted to be a positive case, FN denotes the positive sample predicted to be a negative case, and F1 is the reconciled mean of P and R. The specific results are shown in Table 3.

For the assessment of relevance, this paper first uses a sampling method [33] from the extracted entities to select 10% of the data to ensure that the indicators observed in the sample are within a certain confidence interval and generalized to the overall population at a certain confidence level. Secondly, three undergraduate students majoring in education who have not participated in this experiment are invited, two of them annotate whether the extracted entities are related to the entity types defined in this paper, and the third student makes the final decision when the two results are contradictory, and the Kappa coefficient of the annotations [34] is 0.91, which can be regarded as a basic consistency of the annotation results. Finally, the annotated data is used to train the bert-based classifier, and the detailed results are shown in Table 4.

For the assessment of comprehensiveness, since this paper is based on the construction of the knowledge graph for the graduation requirements of engineering management majors,

**Table 3. Results for each type of triple.**

| Triple Type | P | R | F1 |
|---|---|---|---|
| Graduation Requirements Connotation Observatory—Graduation Requirements | 85.51% | 85.63% | 84.57% |
| Program Objectives—Graduation Requirements Connotation Points | 86.06% | 90.16% | 88.05% |
| Knowledge unit—course objectives | 91.82% | 91.89% | 91.36% |
| Courses—Course Objectives | 84.50% | 87.54% | 87.51% |
| Knowledge units—courses | 90.17% | 90.85% | 89.67% |
| Textbook title—Publisher | 89.12% | 86.85% | 87.79% |
| Textbook Title—Author | 87.18% | 88.78% | 86.56% |
| Courses—Labs | 85.49% | 88.12% | 86.91% |
| Course—Textbook Title | 86.12% | 90.19% | 87.64% |
| Knowledge units—specific knowledge points | 85.19% | 86.21% | 87.18% |
| Courses—Courses | 86.56% | 85.53% | 88.17% |
| average value | 87.07% | 88.34% | 87.76% |

**Table 4. Classification results.**

| Entity type | Accuracy |
|---|---|
| Specific knowledge points | 84.09% |
| knowledge unit | 83.75% |
| Textbook title | 85.65% |
| Course objectives | 83.64% |
| Author | 89.71% |
| Publisher | 88.40% |
| Course title | 89.08% |
| lab | 86.86% |
| Graduation Requirements Connotation Observatory | 81.12% |
| Graduation requirements | 81.87% |
| average value | 85.42% |

and the purpose is to provide reference for other engineering education majors to improve the degree of graduation requirements achievement based on this new paradigm of teaching empowered by the knowledge graph, for the assessment of comprehensiveness, this paper invites six engineering management majors' teachers, who have an average of 12 years of teaching experience, to assess the specific scope covered by the entity types and relationship types as well as the extracted ternary group, for the entity types and relationship types, all five teachers agreed that the entity types and relationship types defined in this paper can cover the actual teaching and learning. For entity types and relationship types, five teachers agree that the entity types and relationship types defined in this paper can cover the contents required in actual teaching and learning, and for the contents of the extracted triad, four teachers think that the contents of the syllabus, teaching design and the related cultivation webpage can be more comprehensive and structured with structured knowledge. The four teachers believe that the contents of the syllabus, teaching design and the knowledge of the related cultivation webpage can be represented in a more comprehensive and structured way, which can provide an intuitive and comprehensive visualized knowledge graph for students.

## Data quality assessment analysis

For the assessment of correctness, from the results shown in Table 3, the mean values of P, R, and F1 reached 87.07%, 88.34%, and 87.76% respectively which can be recognized as high correctness for the extraction of the ternary group. For the assessment of relevance, from the results in Table 4, the scores of Accuracy are above 0.80, indicating that the extracted entities can be considered as the correct entity types, reflecting high relevance. As for the assessment of comprehensiveness, by combining the views of the six teachers, it can be concluded that the data have good comprehensiveness. The above results indicate that the quality of the data is better.

## Use effect evaluation

This paper takes the engineering management major of North China University of Water Resources and Electric Power as an example, and calculates the students' course goal achievement degree by collecting the test scores, experimental scores, and regular scores before and after the use of knowledge graph by the students of the major, and then weights them to get the final graduation requirement achievement degree of each item. The course goal

achievement degree is weighted by the test scores, experimental scores and regular scores, which is the basis of the graduation requirement achievement degree. The specific calculation formula is shown in Eq (5), where A is the usual grade, B is the examination grade, and C is the experimental grade, all of which are in percentage.

$$S = \frac{A * 30\% + B * 50\% + C * 20\%}{100} \tag{5}$$

The degree of graduation requirement achievement is the sum of all the courses that support the graduation requirements. If there are a total of $H$ graduation requirement index points for the nth graduation requirement supported by $i$ courses. Among them, the target attainment of $i$ courses is $S_i$, and the weight of Hu for the nth graduation requirement is $W_x$, then the attainment of $H_u$ for the nth graduation requirement is shown in Eq (6).

$$H_u = \sum W_x S_i \tag{6}$$

The achievement degree $H_n$ of the nth graduation requirement is the sum of the achievement degree $H_u$ of the corresponding index point multiplied by the weight $W_l$, which is calculated as shown in Eq (7).

$$H_n = \sum W_l H_u \tag{7}$$

The graduation requirement attainment degree is the sum of $n$ graduation requirement attainment degrees multiplied by the weight $W_p$, which is calculated as shown in Eq (8).

$$H = \sum W_p H_n \tag{8}$$

## Evaluation and analysis of the effectiveness of use

Based on the calculation of graduation requirement attainment of engineering management students who use knowledge graph and those who do not use knowledge graph as described in this paper, The results of the achievement of all graduation requirements are shown in Fig 8:

According to the results of the above data analysis, students who used knowledge graphs generally outperformed students who did not use knowledge graphs in terms of graduation requirement attainment. The main reason for this phenomenon is that the contents related to graduation requirements are scattered in the text, and it is difficult for students to grasp the connection between knowledge and graduation requirements as a whole, and they are not clear about the leading and following courses or specific knowledge points corresponding to graduation requirements, while the knowledge graph is related, structured and navigable, so students can find out the courses and specific knowledge related to each graduation requirement intuitively and efficiently, and they can navigate to the corresponding contents according to the graph for their weak links, so that students can personalize their own learning and complete graduation requirements efficiently and accurately. The results show that constructing a knowledge graph for correlation between knowledge and graduation requirements can effectively improve students' achievement of graduation requirements, and also prove the necessity and effectiveness of constructing this graph.

## Conclusion and prospect

This study combines the OBE concept with the in-depth analysis of texts such as syllabus, instructional design, and web knowledge about educational contents to obtain the hierarchical

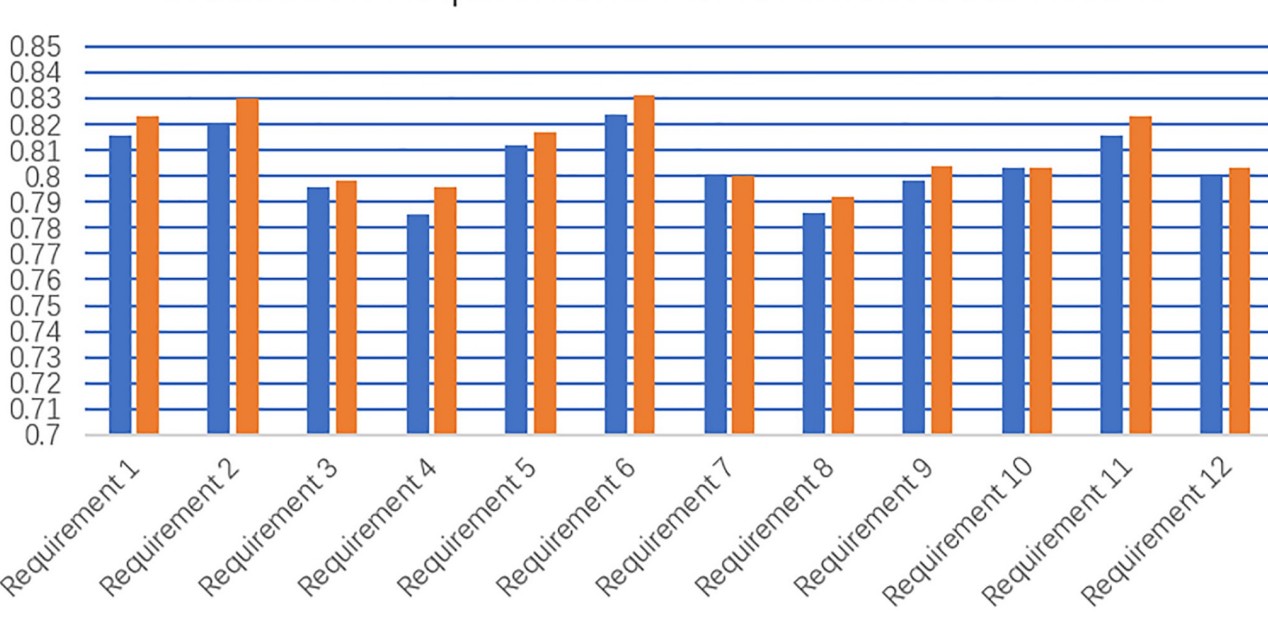

**Fig 8. Results of graduation requirement attainment data.**

structure of "knowledge-course-graduation requirement", and defines 10 types of entities and 11 types of relationships. The ontology is constructed by pre-processing the collected data, and the joint extraction framework EERJE is trained to better extract entity relationship types after using manual annotation of the data, Finally, the content contained in the knowledge graph is also utilized as a neural knowledge base, Allow LLM to learn and then be able to reason about other textual data through the three modules to obtain more entity-relationship triples to further expand the data coverage. Experiments were conducted to assess the quality of the knowledge graph data and the effectiveness of its use, and the results showed that the data quality are better, and the knowledge graph constructed in this way could integrate and visualize the highly dispersed content of graduation requirements, to carefully portray the correlation and difference between knowledge points and graduation requirements, and to build a complete framework, so that students can jump to the corresponding knowledge points according to their own mastery of learning through the contents of the knowledge graph, through this knowledge navigation, so that students can carry out personalized independent learning, strengthen their mastery of knowledge and complete the graduation requirements one step at a time, which improves students' degree of achievement of graduation requirements.

Future work prepares to build a search engine that can support talent development content, assist university teachers in formulating education content with clear objectives and logic, help students complete their professional learning independently, acquire knowledge, skills and professionalism, and lay the foundation for cultivating high-quality talents.

## Author Contributions

**Conceptualization:** Yangrui Yang.

**Methodology:** Yangrui Yang.

**Validation:** Yaping Zhu.

**Visualization:** Yaping Zhu.

**Writing – original draft:** Sisi Chen.

**Writing – review & editing:** Hao Zhu, Zhigang Chen.

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
