## [Decision Letter · Decision Letter 0]

25 Aug 2023

PONE-D-23-17221Knowledge Graph Empowerment from Knowledge Learning to Graduation Requirements AchievementPLOS ONE

Dear Dr. Chen,

Thank you for submitting your manuscript to PLOS ONE. After careful consideration, we feel that it has merit but does not fully meet PLOS ONE’s publication criteria as it currently stands. Therefore, we invite you to submit a revised version of the manuscript that addresses the points raised during the review process.

We look forward to receiving your revised manuscript.

Kind regards,

Sathishkumar V E

Academic Editor

PLOS ONE

Journal Requirements:

"The authors declare no competing interests."

Reviewers' comments:

Reviewer's Responses to Questions

**Comments to the Author**

1. Is the manuscript technically sound, and do the data support the conclusions?

Reviewer #1: Yes

Reviewer #2: Yes

2. Has the statistical analysis been performed appropriately and rigorously? 

Reviewer #1: Yes

Reviewer #2: N/A

3. Have the authors made all data underlying the findings in their manuscript fully available?

Reviewer #1: Yes

Reviewer #2: No

4. Is the manuscript presented in an intelligible fashion and written in standard English?

Reviewer #1: Yes

Reviewer #2: No

5. Review Comments to the Author

Reviewer #1: This paper analyzes the content characteristics of heterogeneous data sources to define clear entity types and inter-entity relationship types. The proposed approach uses EERJE, a joint entity-relationship extraction framework, to perform knowledge triad extraction and build a knowledge graph that supports learning from knowledge toward graduation requirements achievement. The constructed knowledge graph is tested in the teaching and learning process of engineering management majors, and the feedback results show that the graph can effectively help students understand the relationship between knowledge units and graduation requirements.

Discussing state-of-the-art knowledge graph representation learning approaches with contrastive learning would be beneficial for the paper. Contrastive learning has shown promising results in learning high-quality representations for knowledge graphs by leveraging the similarities and differences between entities. Several recent studies have proposed contrastive learning-based methods for knowledge graph representation learning, such as KGCL [1] and CKGC [2].

[1] "Knowledge graph contrastive learning for recommendation", SIGIR'2022

[2] "Cross-modal Knowledge Graph Contrastive Learning for Machine Learning Method Recommendation", ACM MM'2022

Additionally, discussing more details about the parameter settings of compared baselines would also be beneficial for the paper. The performance of machine learning algorithms is highly dependent on the choice of hyperparameters. Therefore, providing more information about the parameter settings used for the compared baselines would enable a fairer and more accurate comparison with the proposed approach.

Reviewer #2: Knowledge graph is a hot topic, it has been applied in different fields. The idea in this paper is interesting and useful, however, I have several suggestions to improve this study:

(1) The abstract needs to be rewritten by reducing some background information but including more specific results.

(2) Why constructing a knowledge graph is necessary for this task, the research objectives, and research questions are unclear. Especially in the era of Generative AI, knowledge graph became less useful. The authors need to argue the importance and the significance of this study.

(3) The authors did not have a good understanding on knowledge graph. There are hundreds of papers from topic conferences and journals studying the knowledge graph construction, evaluation, and applications, however, the author only cite very few references, and most of them are low quality. I believe the authors need to investigate more literature and understand the current methods for knowledge graph construction and evaluation.

(4) The authors need to add some state-of-the-art baseline methods to demonstrate why the proposed method is more effective and useful.

(5) The quality of the knowledge graph should be discussed and evaluated, please cite and refer to the following study:

Chen, H., Cao, G., Chen, J., & Ding, J. (2019). A practical framework for evaluating the quality of knowledge graph. In Knowledge Graph and Semantic Computing: Knowledge Computing and Language Understanding: 4th China Conference, CCKS 2019, Hangzhou, China, August 24–27, 2019, Revised Selected Papers 4 (pp. 111-122). Springer Singapore

(6) I don't think using questionnaire among students to evaluate the knowledge graph makes any sense since human this evaluation could be very subjective. I suggestion the authors to select some evaluation metrics such as correctness, relevance, comprehensive, and others for the evaluation. Also, the authors need to consider developing the downstream application such as recommendation, retrieval, or questions answering to demonstrate the usefulness of the knowledge graph, here is a paper that you can cite and refer:

Nguyen, H., Chen, H., Chen, J., Kargozari, K., & Ding, J. (2023). Construction and evaluation of a domain-specific knowledge graph for knowledge discovery. Information Discovery and Delivery.

(7) The diagrams and figures are low-quality and hard to understand. The writing of the paper is very poor.

6. PLOS authors have the option to publish the peer review history of their article (what does this mean?). If published, this will include your full peer review and any attached files.

Reviewer #1: No

Reviewer #2: **Yes: **Haihua Chen

---

## [Author Response · Author response to Decision Letter 0]

9 Sep 2023

Dear Sathishkumar V E:

On behalf of my co-authors, we thank you very much for giving us an opportunity to revise our manuscript, we appreciate editor and reviewers very much for their positive and constructive comments and suggestions on our manuscript entitled “Knowledge Graph Empowerment from Knowledge Learning to Graduation Requirements Achievement” (ID: PONE-D-23-17221).

In response to additional requests, we provide a detailed explanation of each suggestion:

(1) We have adapted the formatting of the article to meet the style requirements of the plos one journals.

(2) I understand that you expect us to submit code for transparency and reproducibility in the review process. However, due to the fact that intellectual property issues involved in our code, we are unable to publish it publicly here. If we do have the honor of publishing this article, readers can contact the corresponding author and we will determine if we can be of assistance based on the reader's intent..

(3) There are no competing interests between the authors, as we have explained in Competing interests below.

(4) Our data can be found at https://github.com/aichiroudemao/database.

(5) We have converted the pictures in the article into tif format

We apologize for the trouble we have caused you due to our problems, in addition, we have carefully studied the reviewers' comments for revisions and marked them in red and blue in the paper, and we have explained and answered each comment in detail in Response to Reviewers.

We would like to express our great appreciation to you and reviewers for comments on our paper. Looking forward to hearing from you. Thank you and best regards.

Data Availability Statement: Our data can be found here: https://github.com/aichiroudemao/database

Funding:This research was funded by the Henan Provincial Higher Education Teaching Reform Research and Practice Project (2021SJGLX017), and the North China University of Water Resources and Electric Power Education Reform Project.

Competing interests: The authors have declared that no competing interests exist.

Dear Reviewers:

Thanks for your letter and comments concerning our manuscript entitled “Knowledge Graph Empowerment from Knowledge Learning to Graduation Requirements Achievement” (ID: PONE-D-23-17221). Those comments are all valuable and very helpful for revising and improving our paper, as well as the important guiding significance to our researches. We have studied comments carefully and have made correction which we hope meet with approval.

Reviewer #1：

Discussing state-of-the-art knowledge graph representation learning approaches with contrastive learning would be beneficial for the paper. Contrastive learning has shown promising results in learning high-quality representations for knowledge graphs by leveraging the similarities and differences between entities. Several recent studies have proposed contrastive learning-based methods for knowledge graph representation learning, such as KGCL [1] and CKGC [2].

[1] "Knowledge graph contrastive learning for recommendation", SIGIR'2022

[2] "Cross-modal Knowledge Graph Contrastive Learning for Machine Learning Method Recommendation", ACM MM'2022

 Additionally, discussing more details about the parameter settings of compared baselines would also be beneficial for the paper. The performance of machine learning algorithms is highly dependent on the choice of hyperparameters. Therefore, providing more information about the parameter settings used for the compared baselines would enable a fairer and more accurate comparison with the proposed approach.

The author’s answer:

Thank you very much for the advice you've provided us with. Your mention of discussing knowledge graph representation learning methods through contrastive learning methods is helpful for us to improve the quality of the article. Understanding that contrastive learning is an important method in knowledge reasoning, we have made additional improvements to the reasoning section of the article. Specifically, we are using large language models for knowledge reasoning: mainly by letting the knowledge graph constructed in this article, with "knowledge-course-graduation requirements" as the level of neural knowledge base, and letting the large language model learn the knowledge in the graph. We use three modules: entity extraction module, entity knowledge parsing module, and entity relationship decision module to reason about the entity relationship contained in other texts. We have marked the details in blue in the text, Specifically, lines 259-301 in the Knowledge Reasoning section. Regarding the benchmark experiments of inference performed by the Large Language Model, we have validated them in lines 342-394 in the Knowledge Graph Evaluation and Analysis section by evaluating the accuracy, relevance, and comprehensiveness of the data, and the results show that the data are of good quality. Finally, thank you again for your suggestions.

Reviewer #2: 

(1) The abstract needs to be rewritten by reducing some background information but including more specific results.

 (2) Why constructing a knowledge graph is necessary for this task, the research objectives, and research questions are unclear. Especially in the era of Generative AI, knowledge graph became less useful. The authors need to argue the importance and the significance of this study.

 (3) The authors did not have a good understanding on knowledge graph. There are hundreds of papers from topic conferences and journals studying the knowledge graph construction, evaluation, and applications, however, the author only cite very few references, and most of them are low quality. I believe the authors need to investigate more literature and understand the current methods for knowledge graph construction and evaluation.

 (4) The authors need to add some state-of-the-art baseline methods to demonstrate why the proposed method is more effective and useful.

 (5) The quality of the knowledge graph should be discussed and evaluated, please cite and refer to the following study:Chen, H., Cao, G., Chen, J., & Ding, J. (2019). A practical framework for evaluating the quality of knowledge graph. In Knowledge Graph and Semantic Computing: Knowledge Computing and Language Understanding: 4th China Conference, CCKS 2019, Hangzhou, China, August 24–27, 2019, Revised Selected Papers 4 (pp. 111-122). Springer Singapore

 (6) I don't think using questionnaire among students to evaluate the knowledge graph makes any sense since human this evaluation could be very subjective. I suggestion the authors to select some evaluation metrics such as correctness, relevance, comprehensive, and others for the evaluation. Also, the authors need to consider developing the downstream application such as recommendation, retrieval, or questions answering to demonstrate the usefulness of the knowledge graph, here is a paper that you can cite and refer:Nguyen, H., Chen, H., Chen, J., Kargozari, K., & Ding, J. (2023). Construction and evaluation of a domain-specific knowledge graph for knowledge discovery. Information Discovery and Delivery.

 (7) The diagrams and figures are low-quality and hard to understand. The writing of the paper is very poor.

The author’s answer:

Thank you very much for the suggestions you made to us, it was very valuable to us, we have made the changes as you suggested and marked them in red in the article, for the rest of the questions you have asked, I will answer and explain one by one. 

(1) We have rewritten the abstracts to reduce the amount of background information in order to make them more concise and focused on the core of the study. At the same time, we have ensured that more specific findings are included in the abstract to enable the reader to better understand the significance of our findings and discoveries. We hope that these improvements will meet your expectations and further enhance the quality of the abstracts. Thank you again for your valuable comments.

(2) The main objective of our research is to enable students to visualize the connection between knowledge points and graduation requirements, which can be easily appreciated through the structured path of knowledge graph, and can be personalized for self-directed learning based on the content of the knowledge graph. The existing generative AI, on the other hand, despite its great potential in the field of natural language processing, can be used in applications such as automatic text generation, machine translation, dialog systems, intelligent assistants, etc., which can help to improve human-computer interaction and cross-linguistic communication. However, it also has some problems, such as having randomness and easily generating answers that are not related to the question at hand; in terms of vertical domains, it may introduce noise from other domains, and so on. Therefore, considering the above, we choose to construct a knowledge graph to achieve our purpose. In addition, we have added relevant notes in the Abstract section, Lines 108-117 in the Introduction section, Lines 147-151 in the Related Work section, and Lines 425-438 in the Usage Effectiveness Analysis section to emphasize more on the purpose of our study.

(3) Thanks to your suggestion, we have conducted in-depth and extensive literature reading and research to better understand the current methods of knowledge graph construction and evaluation. At the same time, we have supplemented and improved the citations of related literature, such as literature [15]、[17]、[23]、[31]、[32], and we have marked the modified parts in red. In the order in which the literature is cited, specifically in lines 120-121, 124-127, 177, 328-332, 333-337Thanks again for your suggestions.

(4)Thanks to your suggestion, we used the current advanced large language model T5 for knowledge extraction, and objectively assessed the data quality from the perspective of comprehensiveness, relatedness and correctness, and the experimental results proved that the data quality is better, and The knowledge graph constructed using this data can help students intuitively and comprehensively understand the connections and differences between graduation requirements and each knowledge point. It enables students to engage in personalized learning through the navigation provided by the knowledge graph, allowing them to strengthen weak areas, fulfill relevant graduation requirements, and significantly enhance the attainment of graduation criteria. For specific details in lines 341-392 of the Knowledge Graph Assessment and Analysis section Once again, thank you for your suggestions.

(5) Your comments are valuable and applying them to the text will improve the quality of our research. We will cite and refer to relevant research you provide to support our knowledge graph quality assessment. We will explain this in lines 326-396 of the Knowledge Graph Assessment and Analysis section. Thank you very much for your advice.

(6) The suggestions you provided on knowledge graph evaluation methods and downstream applications were very valuable, and we made modifications based on your suggestions to improve the quality and usefulness of our study. Therefore, we drew on the references you provided to assess the objective aspects of this paper's graph, such as correctness, relevance, and comprehensiveness, in order to measure the quality of the knowledge graph and these metrics will help to more accurately assess the validity of our methodology and improve the credibility of the study, which we describe in detail in lines 329-395 in the Knowledge graph Evaluation and Analysis section. At the same time, considering that the purpose of the knowledge graph constructed in this paper is to better fulfill the graduation requirements for school students, investigating whether the knowledge graph can effectively improve the degree of achievement of graduation requirements for school students is equally important to the assessment of the usefulness of the knowledge graph, therefore, this paper adopts a combination of objective and subjective methods to comprehensively assess the quality of the knowledge graph's data and the effectiveness of its use. An assessment of the effectiveness of the use of the knowledge graph, specifically in lines 396-435.In addition, our next step will be to develop downstream applications based on your suggestions and conduct more in-depth research on this work. Thank you again for your suggestions.

(7) In response to the chart issue, we revisited all charts and redesigned them to ensure that they were clearer, better organized, and easier to understand, and used more appropriate color choices, line widths, and font sizes to improve visualization. Every effort was made to improve the manuscript in response to the writing problems, and some changes were made to the manuscript. These changes do not affect the content or framework of the paper. We have highlighted these changes in red in the revised document.We would like to express our sincere thanks for your enthusiastic work and hope that the revised paper will be approved by you.

---

## [Decision Letter · Decision Letter 1]

2 Oct 2023

Knowledge Graph Empowerment from Knowledge Learning to Graduation Requirements Achievement

PONE-D-23-17221R1

Dear Dr. Chen,

We’re pleased to inform you that your manuscript has been judged scientifically suitable for publication and will be formally accepted for publication once it meets all outstanding technical requirements.

Kind regards,

Sathishkumar Veerappampalayam Easwaramoorthy

Academic Editor

PLOS ONE

Additional Editor Comments (optional):

Reviewers' comments:

Reviewer's Responses to Questions

**Comments to the Author**

1. If the authors have adequately addressed your comments raised in a previous round of review and you feel that this manuscript is now acceptable for publication, you may indicate that here to bypass the “Comments to the Author” section, enter your conflict of interest statement in the “Confidential to Editor” section, and submit your "Accept" recommendation.

Reviewer #1: All comments have been addressed

2. Is the manuscript technically sound, and do the data support the conclusions?

Reviewer #1: Yes

3. Has the statistical analysis been performed appropriately and rigorously? 

Reviewer #1: Yes

4. Have the authors made all data underlying the findings in their manuscript fully available?

Reviewer #1: Yes

5. Is the manuscript presented in an intelligible fashion and written in standard English?

Reviewer #1: Yes

6. Review Comments to the Author

Reviewer #1: My comments on this work have been addressed. I have no further comment and recommend accept this paper

7. PLOS authors have the option to publish the peer review history of their article (what does this mean?). If published, this will include your full peer review and any attached files.

Reviewer #1: **Yes: **Chao Huang

---

## [Editor Report · Acceptance letter]

4 Oct 2023

PONE-D-23-17221R1 

Knowledge Graph Empowerment from Knowledge Learning to Graduation Requirements Achievement 

Dear Dr. Chen:

I'm pleased to inform you that your manuscript has been deemed suitable for publication in PLOS ONE. Congratulations! Your manuscript is now with our production department. 

Kind regards, 

on behalf of

Dr. Sathishkumar Veerappampalayam Easwaramoorthy 

Academic Editor

PLOS ONE